# Revisiting Continuity of Image Tokens for Cross-Domain Few-shot Learning

Shuai Yi [1 2]   Yixiong Zou[✉ 1]   Yuhua Li[✉ 1]   Ruixuan Li[✉ 1]

## Abstract

Vision Transformer (ViT) has achieved remarkable success due to its large-scale pretraining on general domains, but it still faces challenges when applying it to downstream distant domains that have only scarce training data, which gives rise to the Cross-Domain Few-Shot Learning (CDFSL) task. Inspired by Self-Attention's insensitivity to token orders, we find an interesting phenomenon neglected in current works: disrupting the continuity of image tokens (i.e., making pixels not smoothly transited across patches) in ViT leads to a noticeable performance decline in the general (source) domain but only a marginal decrease in downstream target domains. This questions the role of image tokens' continuity in ViT's generalization under large domain gaps. In this paper, we delve into this phenomenon for an interpretation. We find continuity aids ViT in learning larger spatial patterns, which are harder to transfer than smaller ones, enlarging domain distances. Meanwhile, it implies that only smaller patterns within each patch could be transferred under extreme domain gaps. Based on this interpretation, we further propose a simple yet effective method for CDFSL that better disrupts the continuity of image tokens, encouraging the model to rely less on large patterns and more on smaller ones. Extensive experiments show the effectiveness of our method in reducing domain gaps and outperforming state-of-the-art works. Codes and models are available at https://github.com/shuaiyi308/ReCIT.

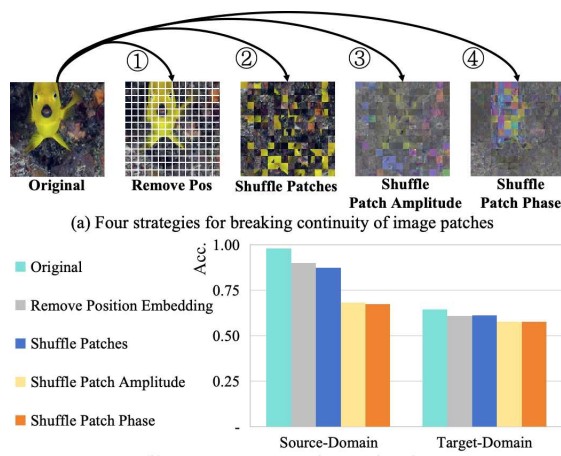

*Figure 1.* (a) Four approaches are utilized to disrupt the continuity of image tokens, i.e., making the pixels not smoothly transited across patches. (b) We find an interesting phenomenon: although disrupting the continuity of image tokens in the source domain has a substantial impact on the performance of ViT-based models, the model's performance in the target domain, which undergoes an equivalent level of continuity disruption, is only marginally affected. In this paper, we will delve into this phenomenon for an interpretation, explore the role of image tokens' continuity in model generalization, and propose methods based on it for better cross-domain few-shot learning.

## 1. Introduction

Vision Transformer (ViT) has achieved great success across numerous tasks (Yuan et al., 2021; Wu et al., 2022) because of its ability to learn from large-scale datasets (Naseer et al., 2021), which makes it a prevailing option for downstream applications by generalizing an upstream-pretrained ViT to downstream expert tasks. However, in real-world scenarios, downstream tasks can be in domains distant from upstream large-scale datasets, and it may not be easy for downstream tasks to collect sufficient training samples, which makes it challenging for the generalization and finetuning of ViT (Zou et al., 2022; 2024b). To mitigate this issue, Cross-Domain Few-Shot Learning (CDFSL) has been proposed, aiming to transfer general knowledge from source domains, such as ImageNet (Deng et al., 2009), to target domains, like medical datasets (Mohanty et al., 2016), that possess only a scarcity of training samples (Oh et al., 2022).

However, the generalization of ViT under large domain

[1]School of Computer Science and Technology, Huazhong University of Science and Technology, Wuhan, China [2]School of Artificial Intelligence and Automation, Huazhong University of Science and Technology, Wuhan, China. Correspondence to: Yixiong Zou <yixiongz@hust.edu.cn>, Yuhua Li <idcliyuhua@hust.edu.cn>, Ruixuan Li <rxli@hust.edu.cn>.

*Proceedings of the 42nd International Conference on Machine Learning*, Vancouver, Canada. PMLR 267, 2025. Copyright 2025 by the author(s).

gaps is still under-explored. Different from Convolutional Neural Networks (CNN), ViT partitions the image into non-overlapping patches for input and takes Multi-head Self-Attention (MSA) (Chen et al., 2023) to model the coherence of patches. However, MSA itself is not sensitive to the order of input tokens, i.e., two subsequent tokens do not have to be continuous in pixels. The only assurance of continuity is the positional embeddings (Dosovitskiy et al., 2021), which is relatively weak. Inspired by this characteristic, we find an intriguing phenomenon that is ignored by current works: when the continuity of images is disrupted e.g., by removing positional embeddings, shuffling image patches, or shuffling the amplitude or phase of image patches in the frequency domain (Fig. 1a), the performance of ViT experiences a noticeable decline in the source domain but decreases only marginally on target domains (Fig. 1b). This phenomenon questions the role of image tokens' continuity in ViT's generalization under large domain gaps.

In this paper, we delve into this phenomenon for an interpretation. We discover that disrupting image tokens' continuity can paradoxically be beneficial in reducing domain gaps. Then, we find that the more disrupted image tokens' continuity (by binding fewer patches not disrupted), the more generally increased domain similarity between source and target domains. Based on these experiments, we interpret the continuity as an aid of ViT in learning larger spatial patterns. However, since large spatial patterns, e.g., a whole dog, are harder to transfer to target domains than smaller patterns, e.g., the head of a dog, under extremely large domain gaps, this phenomenon implies that only smaller patterns within each patch could be transferred to target domains, therefore interrupting image tokens' continuity has only marginal effect on target-domain performance.

Drawing upon this interpretation, we further propose a simple but effective method tailored for the CDFSL task that better disrupts the continuity of image tokens, encouraging the model to strengthen the learning of smaller spatial patterns and reduce its reliance on large ones, thereby enhancing the model's generalization to downstream tasks. Specifically, we integrate the continuity disruption of image tokens in both spatial and frequency domains, and construct a balanced disruption among different style distributions, ensuring the diversity of the disruption. Extensive experiments on four CDFSL benchmarks with large domain gaps show that we can outperform state-of-the-art performance and effectively reduce domain gaps.

In summary, our contributions can be listed as follows.

- To the best of our knowledge, we are the first to consider image tokens' continuity in the CDFSL task.

- We find a phenomenon that is neglected by others: pretrained ViT undergoes a much less performance decline when disrupting image tokens' continuity on target domains than on general (source) domains.

- We delve into this phenomenon for an interpretation: continuity aids ViT in learning large spatial patterns; however, under large domain gaps, large patterns across patches are hard to transfer, making the disruption of large patterns less effective on target domains.

- Based on this interpretation, we further propose a novel method to better disrupt the continuity of image tokens for CDFSL, thereby enabling the model to reduce its reliance on large patterns and enhance its learning of smaller ones, thereby enhancing its transferability.

- Extensive experiments on four benchmark datasets validate our rationale and state-of-the-art performance.

## 2. Delve into Continuity of Image Tokens in Cross-Domain Few-Shot Learning

### 2.1. Preliminaries

Cross-Domain Few-Shot Learning (CDFSL) entails a model to acquire knowledge from a source-domain dataset abundant with training samples (e.g., *mini*ImageNet (Vinyals et al., 2016)), then transfer it to downstream tasks, enabling learning of target-domain datasets using merely a handful of training instances. Finally, the model is evaluated on target datasets.

Specifically, we denote the source dataset as $D^S = \{x_i^S, y_i^S\}_{i=1}^N$ with $x_i^S$ and $y_i^S$ symbolizing the $i$th training sample and its corresponding label, respectively. Analogously, $D^T = \{x_i^T, y_i^T\}_{i=1}^{N'}$ represents the target dataset. During the learning and evaluation phases on $D^T$, to ensure a fair comparison, current research (Fu et al., 2022; Zou et al., b) employs a $k$-way $n$-shot paradigm. This involves sampling from $D^T$ to construct limited datasets, known as episodes, each comprising $k$ classes with $n$ training samples per class. Based on these episodes, the model learns from the $k * n$ samples, collectively termed the support set $\{x_{ij}^T, y_{ij}^T\}_{i=1,j=1}^{k,n}$, and its performance is assessed using testing samples from the same $k$ classes, referred to as the query set $\{x_q^T\}$.

The Vision Transformer (ViT) has recently gained significant popularity in vision-related tasks. It operates by dividing an image $x \in R^{H \times W \times C}$ into fixed-sized patches $x_p \in R^{M \times (P^2 \cdot C)}$, Where $(H, W)$ denotes the resolution of the original image, $C$ represents the number of channels, $(P, P)$ signifies the resolution of each image patch, and $M = HW/P^2$ is the resulting count of patches. This count, $M$, also functions as the number of input tokens for the Transformer. Then each image patch is flattened and projected into a D-dimensional space through a trainable

linear projection $E \in R^{(P^2 \cdot C) \times D}$ termed patch embeddings. Additionally, a learnable embedding called $class$ token (denoted as $x_{class}$) is prepended to the sequence of patch embeddings. To maintain positional information, the patch embeddings are added with position embeddings $E_{\text{pos}} \in R^{(M+1) \times D}$, which can be represented as

$$z_0 = \left[ x_{\text{class}}; x_p^1 E; x_p^2 E; \cdots; x_p^M E \right] + E_{\text{pos}}, \quad (1)$$

The resultant sequence of embedding vectors subsequently serves as the input to an encoder architecture that consists of $L$ stacked blocks, each encompassing a multiheaded self-attention (MSA) network, a Multi-Layer Perceptron (MLP) network, Layernorm (LN) and residual connections. The overall process can be depicted as follows

$$z_l' = \text{MSA} \left( \text{LN} \left( z_{l-1} \right) \right) + z_{l-1}, \quad (2)$$
$$z_l = \text{MLP} \left( \text{LN} \left( z_l' \right) \right) + z_l', \quad (3)$$
$$f(x_i^S) = \text{LN} \left( z_L \right), \quad (4)$$

In this paper, we focus on exploring ViT's downstream generalization on the CDFSL task. We follow (Fu et al., 2023; Zou et al., a) to employ the DINO (Zhang et al., 2022) pretraining on ImageNet (Deng et al., 2009) as the initialization. Then, we train the ViT on $D^S$ by minimizing the cross-entropy loss relevant to the source-domain label space $|Y^S|$, with a fully connected (FC) layer as

$$L = \frac{1}{N} \sum_i^N L_{cls}(\phi(f(x_i^S)), y_i^S), \quad (5)$$

where $\phi(\cdot)$ represents the FC layer and $f(\cdot)$ denotes ViT. Finally, we employ prototype-based classification (Zhou et al., 2023) for target-domain recognition with a distance function $d(\cdot, \cdot)$ as

$$\hat{y}_q^T = \arg\min_i d(\frac{1}{n} \sum_j f(x_{ij}^T), f(x_q^T)), \quad (6)$$

### 2.2. Breaking the continuity of image tokens

As illustrated in Fig. 1a, we first contemplate four of the simplest approaches to disrupt the continuity of images, applying each separately to the training set within the source domain to facilitate the model's training on that domain.

**Remove Position Embedding (RPE)**: The first method consists of inputting the image patches directly into the encoder of ViT, without incorporating positional embeddings as

$$z_0 = \left[ x_{\text{class}}; x_p^1 E; x_p^2 E; \cdots; x_p^M E \right], \quad (7)$$

**Shuffle Patches (SP)**: The second approach involves shuffling the image patches directly, concatenating them with a

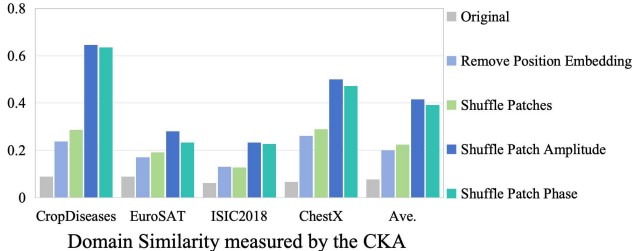

*Figure 2.* Disrupting the continuity of input images significantly increases domain similarity, although the performance decreases on all datasets in Fig. 1. Intriguingly, the more decrease in Fig. 1, the more increase in domain similarity, which shows disrupting continuity can surprisingly mitigate domain discrepancies.

class token, and then adding positional embeddings before inputting them into the encoder:

$$z_0 = \left[ x_{\text{class}}; x_p^{1'} E; x_p^{2'} E; \cdots; x_p^{M'} E \right] + E_{\text{pos}}, \quad (8)$$

**Shuffle Patch Amplitude (SPA)**: The two subsequent methods involve transitioning image patches from the spatial domain to the frequency domain, starting with obtaining the Fourier transformations of the input patches $x_p$

$$F(x_p)[m, n] = \sum_{h=0}^{H'-1} \sum_{w=0}^{W'-1} x_p[h, w] \exp\left(-2\pi\left(\frac{h}{H'}m + \frac{w}{W'}n\right)\right), \quad (9)$$

where $i^2 = -1$ and $m, n$ denote spatial frequencies. When $Re(F(x)[\cdot, \cdot])$ and $Im(F(x)[\cdot, \cdot])$ represent the real and imaginary components of the Fourier spectrum, respectively, the corresponding amplitude spectrum $A(x)[\cdot, \cdot]$ and phase spectrum $P(x)[\cdot, \cdot]$ can be expressed as follows

$$A(x_p)[m, n] = \sqrt{Re(F(x_p)[m, n])^2 + Im(F(x)[m, n])^2}, \quad (10)$$

$$P(x_p)[m, n] = \arctan\left(\frac{Im(F(x_p)[m, n])}{Re(F(x_p)[m, n])}\right), \quad (11)$$

Our third method disrupts the amplitudes of the patches within an image, merges them with the original phases of these patches, and then applies an inverse Fourier transform to revert the combined data back to the spatial domain as

$$x_p^k = \text{iDFT}\left(A_{x_p^{k'}} \otimes e^{i \cdot P_{x_p^k}}\right), \quad (12)$$

where $k$ denote $k$th patch in image, $k'$ denote after be shuffed, the $k$th patch's amplitude.

**Shuffle Patch Phase (SPP)**: The fourth method, in contrast to the third one, disrupts continuity by shuffling the phases while retaining the amplitudes as

$$x_p^k = \text{iDFT}\left(A_{x_p^k} \otimes e^{i \cdot P_{x_p^{k'}}}\right), \quad (13)$$

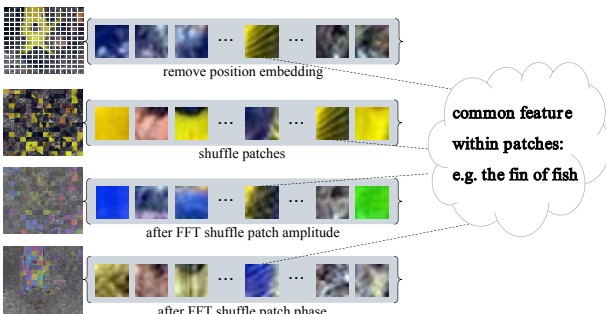

*Figure 3.* Take a fish as an example, although disrupting continuity distorts its overall shape, it is still feasible to recognize the fish's patterns in individual patches, such as fins and eyes. This indicates that the continuity between patches primarily assists the model in learning larger spatial patterns; however, even after disrupting the continuity, the model can only recognize the patterns maintained within each patch, which is smaller but easier to transfer.

## 2.3. What role does ViT's weak continuity play under large domain gaps?

In Fig. 1, we observe that a pretrained ViT is highly sensitive to disruptions in continuity on the source domain that is similar to its pretraining data, yet exhibits a much smaller impact on target domains that are distant from the source domain. Therefore, we are inspired to question the role that image tokens' continuity plays under large domain gaps.

Firstly, we quantitatively assess the domain distance between the source and target domains using the Centered Kernel Alignment (CKA (Kornblith et al., 2019)) similarity following (Oh et al., 2022; Davari et al., 2022). Specifically, utilizing ViT as the backbone network, we extract features from images belonging to different domains and then compute the CKA similarity between different domains' features by aligning the channel dimension. A higher CKA similarity indicates a smaller domain distance, implying that the model encompasses less domain-specific information.

The results are in Fig. 2, from which we observe the domain similarity between the source and target datasets increased significantly, although in Fig. 1 the model's performance declines on all datasets. Intriguingly, the greater the decline in model performance in Fig. 1, the more the increase in domain similarities. This indicates **the surprising benefit of disrupting continuity in reducing domain discrepancies**, thereby enhancing the model's transferability.

## 2.4. Why does breaking continuity reduce domain discrepancy?

To account for breaking continuity reduces domain distance, we look back on the disrupted images. Although the connection of each patch is disrupted, patterns within each patch are not disrupted. For example, as shown in Fig. 3, given

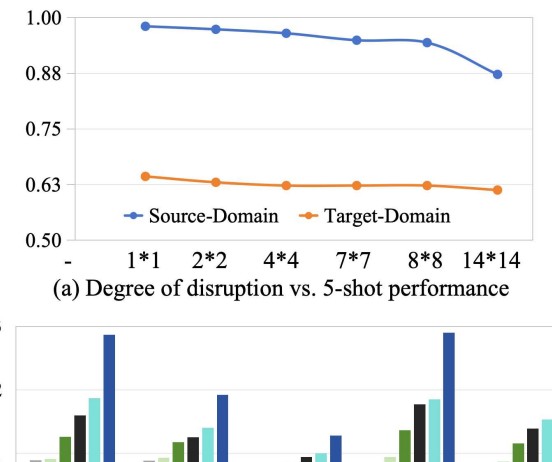

(a) Degree of disruption vs. 5-shot performance

(b)Domain Similarity measured by the CKA

*Figure 4.* We sequentially divide the images into pseudo-patches, where the pseudo-patch size decreases from left to right, and shuffle them. The smaller the patterns preserved within these pseudo-patches, (a) the more the model's performance decreases, and (b) the more the domain similarity increases. This suggests that breaking the continuity essentially influences the spatial size of maintained patterns, where larger ones are discriminative on the source domain but harder to transfer, and vice versa.

an image of a fish, although the patches are shuffled, the internal details of each patch still allow us to discern features such as fish scales, eyes, fins, tail, and other distinctive attributes. Based on such a disrupted image, the model extracts features majorly based on the patterns maintained within each patch, while the patterns across patches are lost.

Intuitively, patterns within each patch are spatially smaller, such as fish eyes, while patches are interconnected with their neighbors to build spatially larger patterns, such as a whole fish. Simultaneously, larger patterns are always harder to transfer than smaller ones (Zou et al., 2024b). For example, capturing a pattern similar to a whole fish in a dog is difficult, but capturing a pattern similar to a fish eye is relatively easier, e.g., the dog eyes. Therefore, we hypothesize that **breaking continuity essentially breaks the large patterns across patches into small patterns within each patch**. Under large domain gaps, large patterns are inherently harder to transfer. Therefore, the major effective patterns on target domains are small patterns within each patch. Consequently, *disrupting token continuity has a much smaller effect on target domains, as the recognition of target domains is only marginally based on large patterns across patches*. Similarly, since larger patterns are effective for the source domain, the performance of source-domain recognition drops drastically. On the other hand, breaking

continuity forces the model to extract features majorly based on smaller patterns, aligning with the patterns used on target domains. Therefore, the domain similarity increases.

To validate our hypothesis, we disrupt images to varying degrees of shuffling, in order to verify how the maintained patterns' spatial size influences the performance and domain similarity. Specifically, each image is divided into pseudo-patches of sizes $1 * 1$, $2 * 2$, $4 * 4$, $7 * 7$, $8 * 8$, and $14 * 14$. For instance, $2 * 2$ signifies dividing the whole image into four equal-sized sections along both its length and width[1]. Then, these pseudo-patches are randomly shuffled and then reassembled to form the shuffled images, which are then input into ViT. Since patches within each pseudo-patch are not disrupted, the size of spatial patterns is larger.

As in Fig. 4a, the performance consistently decreases with the growing number of pseudo-patches, i.e., smaller patterns within each pseudo-patch. Meanwhile, domain similarities between source and target domains consistently increase as the pseudo-patch size reduces, verifying it is the maintained pattern's spatial size that influences domain similarities.

## 2.5. Conclusion and Discussion

Based on it, we interpret as follows. The continuity between image tokens essentially assists models in learning larger spatial patterns, which are beneficial for source-domain classification, therefore breaking the continuity significantly harms source-domain performance. However, excessively large patterns often struggle to transfer to target domains. Conversely, smaller patterns are easier to transfer. Under large domain gaps, most patterns transferred to target domains are those small ones that are within each patch, therefore disrupting the continuity has smaller influences on the target-domain performance. By aligning patterns used in feature extraction for source and target domains to small patterns, the domain similarity also increases.

## 3. Method

Building upon the above analysis and interpretation, we conclude that under large domain gaps, maybe only small patterns within each patch can be transferred. Therefore, we aim to prioritize learning within patches and reduce models' reliance on large patterns across patches. Specifically, during the source-domain training, we propose to disrupt the continuity of image tokens in two steps, including a warm-up spatial-domain disruption and a balanced frequency-domain disruption (Fig. 5).

---

[1]We resize the image to $256 * 256$ if the number of tokens is not evenly divided, preserving the number of pixels in each patch.

## 3.1. Warm-Up Spatial-Domain Disruption

Since the model's pretraining data is significantly different from the disrupted data, the training on the disrupted ones may be difficult. Therefore, gradually increasing the difficulty will be beneficial for model training. As illustrated in Fig. 4b, the magnitude of Shuffle Patches' performance drop and domain-similarity increase is smaller than that of the amplitude-based disruption. Therefore, we first use Shuffle Patches as a spatial-domain disruption for warming up.

Specifically, our method involves randomly dividing images into varying numbers of patches and subsequently shuffling these patches. This approach encourages the model to learn the internal information within patches of different sizes, gradually enabling it to adapt and learn smaller patterns. As in the Fig. 5①, we divide a set of images into a random number of equally sized patches (e.g., $L$ patches). Afterward, we shuffle the patches belonging to the same image, append the CLS token and positional encoding to them, and then feed them into the ViT as

$$z_0 = \left[ x_{\text{class}}; x_p^{1'} E; x_p^{2'} E; \cdots ; x_p^{L'} E \right] + E_{\text{pos}}, \quad (14)$$

## 3.2. Balanced Frequency-Domain Disruption

Considering the comparison of the four simple continuity disruption methods presented in Fig. 2, it is observed that shuffling the amplitudes in the frequency domain maximizes domain similarity between the source and target domains. This consideration is primarily based on the following two reasons. Firstly, transferred patterns may also encompass regions in adjacent patches, e.g., a fish eye split into two adjacent patches, suggesting that retaining a certain degree of continuity is beneficial. Secondly, studies (Chen et al., 2021) have demonstrated that some information in the frequency domain encompasses domain information. By shuffling the amplitudes, the model can eliminate style biases. Therefore, we aim to take Shuffle Patch Amplitude as a strong frequency-domain disruption to enhance the learning of small and domain-agnostic patterns.

Considering the distribution of patches may vary across images, simply shuffling the amplitudes of image patches directly may not diversely disrupt the continuity. Take the classification of jellyfish in the ocean as an example: most patches may be dark water. When we directly shuffle the amplitudes of image patches, the overall image remains largely unchanged, with most areas retaining the continuity. Therefore, we consider balancing the distribution of patches for better disruption of the continuity.

Therefore, we first cluster image patches based on their visual appearances, randomly sample styles from each cluster, and finally re-assign the sampled styles to each patch. This approach balances different sizes of image clusters, e.g.,

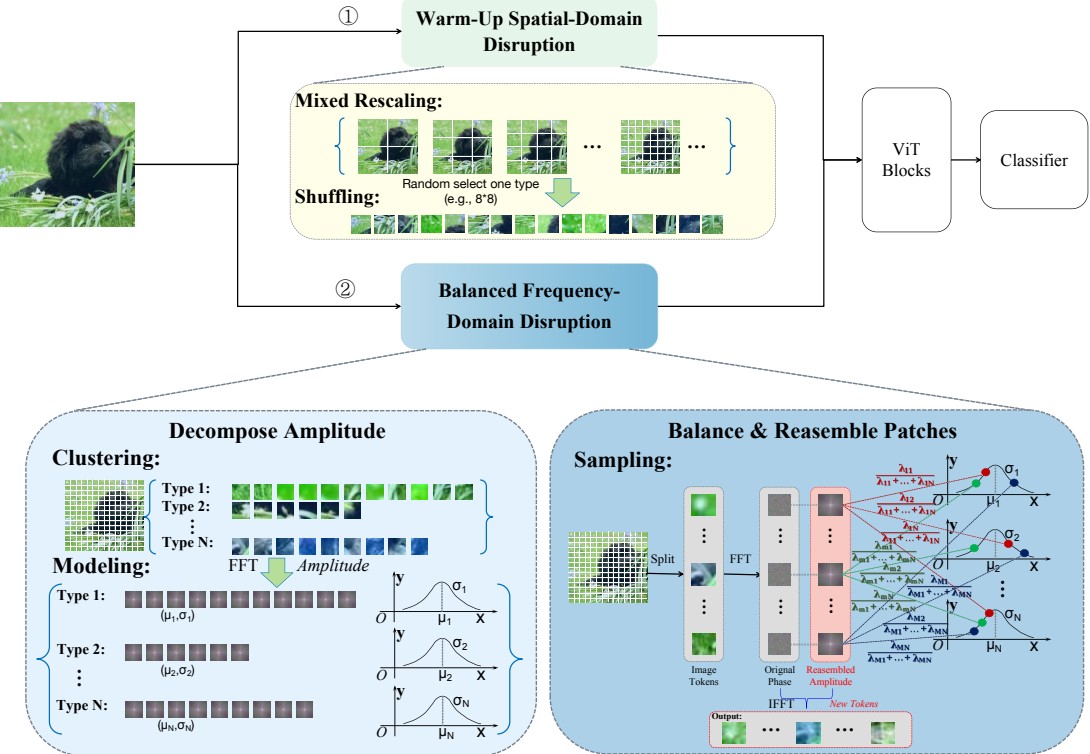

*Figure 5.* We take two steps to disrupt the continuity of images during source-domain training. Step ① involves dividing the image into a random number of patches and then shuffling them. This approach poses a relatively lower difficulty for the model and is utilized during the warming up of training. Step ② decomposes and reassembles the amplitudes of the clustered image patches to ensure the diversity of disruptions, which presents a greater challenge for the model and is thus employed during the middle to later stages of training.

even if dark water takes a large area in the jellyfish image, its sampling probability is similar to those in small areas.

Specifically, as in Fig. 5②, an input image $x \in R^{H \times W \times C}$ is split into the patches $x_p \in R^{M \times D}$ where $x_p^i$ and $x_p^j$ are the average of pixels in the $i$th and $j$th patches ($i, j \in M$). Then, we calculate the similarity distance among these patches by employing the cosine similarity metric

$$cos(x_p^i, x_p^j) = \frac{x_p^i \cdot x_p^j}{||x_p^i|| \cdot ||x_p^j||}, \quad (15)$$

Subsequently, we categorize these patches into different types based on their similarity exceeding a predefined threshold, thereby dividing all patches within an image into multiple clusters. A cluster is defined as comprising a patch $x_p^i$ and its corresponding patches, all of which fall within a predefined similarity threshold $sim$

$$Cluster_{x_p^i} = \{x_p^j \mid cos(x_p^i, x_p^j) \geq sim, x_p^j \in x_p\}, \quad (16)$$

Next, as shown in the Eq. 10, we extract the amplitudes of each patch within each cluster

$$Cluster_{A_p^i} = \{A_p^j \mid cos(x_p^i, x_p^j) \geq sim, x_p^j \in x_p\}, \quad (17)$$

where $A_p^j$ is the amplitude corresponding to the patch $x_p^j$, $M_i$ is the total numbers of patches in this cluster.

Then, we model the amplitude distribution of each patch within each cluster as a multivariate Gaussian distribution. This Gaussian distribution centers around the mean amplitude of all patches within the corresponding cluster and its variance can be computed from the differences between the amplitudes of the patches within the cluster and their mean value, thereby obtaining a Gaussian distribution that represents that particular amplitude of cluster.

$$\mu(Cluster_{A_p^i}) = \frac{1}{M_i} \sum_{j=1}^{M_i} A_p^j, \quad A_p^j \in Cluster_{A_p^i}, \quad (18)$$

$$\sigma(Cluster_{A_p^i}) = \frac{1}{M_i} \sum_{j=1}^{M_i} \left[ A_p^j - \mu(Cluster_{A_p^i}) \right]^2, \quad (19)$$

After extracting the distributions of multiple amplitudes from an image, we random sample on each cluster distribution, e.g. $\epsilon_{Cluster_{A_p^i}} \sim N(\mu(Cluster_{A_p^i}), \sigma(Cluster_{A_p^i}))$

Finally, we sum these samples with random proportions $p$ to

*Table 1.* Comparison with state-of-the-art works based on ViT-S on target domains.

| Method | Shot | FT | Mark | ChestX | ISIC2018 | EuroSAT | CropDiseases | Average |
|---|---|---|---|---|---|---|---|---|
| MEM-FS (Walsh et al., 2023) | 1 | × | TIP-23 | 22.76 | 32.97 | 68.11 | 81.11 | 51.24 |
| StyleAdv (Fu et al., 2023) | 1 | × | CVPR-23 | 22.92 | 33.05 | 72.15 | 81.22 | 52.34 |
| FLoR (Zou et al., 2024a) | 1 | × | CVPR-24 | 22.78 | 34.20 | 72.39 | 81.81 | 52.80 |
| DAMIM (Ma et al., 2024) | 1 | × | AAAI-25 | 22.97 | 34.66 | 72.87 | 82.34 | 53.21 |
| CD-CLS (Zou et al., b) | 1 | × | NeurIPS-24 | 22.93 | 34.21 | 74.08 | 83.51 | 53.68 |
| AttnTemp (Zou et al., a) | 1 | × | NeurIPS-24 | 23.19 | 34.92 | 74.35 | 84.02 | 54.12 |
| **ReCIT** | 1 | × | **Ours** | **23.27** | **35.13** | **74.56** | **84.76** | **54.43** |
| PMF (Shell Xu, 2022) | 1 | ✓ | CVPR-22 | 21.73 | 30.36 | 70.74 | 80.79 | 50.91 |
| FLoR (Zou et al., 2024a) | 1 | ✓ | CVPR-24 | 23.26 | 35.49 | 73.09 | 83.55 | 53.85 |
| StyleAdv (Fu et al., 2023) | 1 | ✓ | CVPR-23 | 22.92 | 33.99 | 74.93 | 84.11 | 53.99 |
| DAMIM (Ma et al., 2024) | 1 | ✓ | AAAI-25 | 23.38 | 36.35 | 73.61 | 83.90 | 54.31 |
| CD-CLS (Zou et al., b) | 1 | ✓ | NeurIPS-24 | 23.39 | 35.56 | 74.97 | 84.54 | 54.62 |
| AttnTemp (Zou et al., a) | 1 | ✓ | NeurIPS-24 | 23.63 | 38.05 | 75.09 | 84.78 | 55.39 |
| **ReCIT** | 1 | ✓ | **Ours** | **23.84** | **38.48** | **75.23** | **85.92** | **55.87** |
| MEM-FS + RDA[*] (Walsh et al., 2023) | 1 | ✓ | TIP-23 | 23.85 | 37.07 | 75.91 | 83.74 | 55.14 |
| DAMIM[*] (Ma et al., 2024) | 1 | ✓ | AAAI-25 | 23.91 | 38.07 | 77.23 | 86.74 | 56.49 |
| CD-CLS (Zou et al., b) | 1 | ✓ | NeurIPS-24 | 23.88 | 37.20 | 78.41 | 87.39 | 56.72 |
| AttnTemp (Zou et al., a) | 1 | ✓ | NeurIPS-24 | 23.96 | **40.13** | 77.40 | 87.58 | 57.23 |
| **ReCIT[*]** | 1 | ✓ | **Ours** | **24.42** | 39.39 | **78.84** | **88.45** | **57.78** |
| MEM-FS (Walsh et al., 2023) | 5 | × | TIP-23 | 26.67 | 47.38 | 86.49 | 93.74 | 63.57 |
| StyleAdv (Fu et al., 2023) | 5 | × | CVPR-23 | 26.97 | 47.73 | 88.57 | 94.85 | 64.53 |
| FLoR (Zou et al., 2024a) | 5 | × | CVPR-24 | 26.71 | 49.52 | 90.41 | 95.28 | 65.48 |
| DAMIM (Ma et al., 2024) | 5 | × | AAAI-25 | 27.28 | 50.76 | 89.50 | 95.52 | 65.77 |
| CD-CLS (Zou et al., b) | 5 | × | NeurIPS-24 | 27.23 | 50.46 | **91.04** | 95.68 | 66.10 |
| AttnTemp (Zou et al., a) | 5 | × | NeurIPS-24 | 27.72 | **53.09** | 90.13 | 95.53 | 66.62 |
| **ReCIT** | 5 | × | **Ours** | **28.23** | 52.36 | 90.42 | **96.02** | **66.76** |
| PMF (Shell Xu, 2022) | 5 | ✓ | CVPR-22 | 27.27 | 50.12 | 85.98 | 92.96 | 64.08 |
| StyleAdv (Fu et al., 2023) | 5 | ✓ | CVPR-23 | 26.97 | 51.23 | 90.12 | 95.99 | 66.08 |
| FLoR (Zou et al., 2024a) | 5 | ✓ | CVPR-24 | 27.02 | 53.06 | 90.75 | 96.47 | 66.83 |
| DAMIM (Ma et al., 2024) | 5 | ✓ | AAAI-25 | 27.82 | 54.86 | 91.18 | 96.34 | 67.55 |
| CD-CLS (Zou et al., b) | 5 | ✓ | NeurIPS-24 | 27.66 | 54.69 | 91.53 | 96.27 | 67.54 |
| AttnTemp (Zou et al., a) | 5 | ✓ | NeurIPS-24 | 28.03 | 54.91 | 90.82 | 96.66 | 67.61 |
| **ReCIT** | 5 | ✓ | **Ours** | **28.88** | **54.91** | **91.58** | **96.85** | **68.06** |
| MEM-FS + RDA[*] (Walsh et al., 2023) | 5 | ✓ | TIP-23 | 27.98 | 51.02 | 88.77 | 95.04 | 65.70 |
| DAMIM[*] (Ma et al., 2024) | 5 | ✓ | AAAI-25 | 28.10 | 55.44 | 91.08 | 96.49 | 67.78 |
| CD-CLS[*] (Zou et al., b) | 5 | ✓ | NeurIPS-24 | 28.25 | **55.66** | 91.68 | 96.62 | 68.05 |
| AttnTemp[*] (Zou et al., a) | 5 | ✓ | NeurIPS-24 | 28.41 | 55.22 | 91.34 | 96.74 | 67.93 |
| **ReCIT[*]** | 5 | ✓ | **Ours** | **28.97** | 55.60 | **91.72** | **96.97** | **68.32** |

assign to patches, $N$ is the number of clusters in an image

$$A_p^j = \sum_{i=1}^{N} p_{A_p^i} * \epsilon_{Cluster_{A_p^i}}, \quad (20)$$

$$p_{A_p^i} = \frac{\epsilon_{pro_{A_p^i}}}{\sum_{i=1}^{N} \epsilon_{pro_{A_p^i}}}, \epsilon_{pro_{A_p^i}} \sim N(0, \alpha) \quad (21)$$

This process reconfigures the amplitudes of each patch, addressing the issue of uneven patch distribution within the image and ensuring continuity in amplitudes among patches.

During the target-domain stage, we conduct prototype-based classification (Eq. 6) or finetune-based classification following (Zhou et al., 2023).

## 4. Experiments

### 4.1. Dataset and Implementation Details

Following current works (Oh et al., 2022), we employ the *mini*ImageNet dataset (Vinyals et al., 2016) as our source domain, and target domains involve four datasets: CropDisease (Mohanty et al., 2016), EuroSAT (Helber et al., 2019), ISIC (Codella et al., 2019), and ChestX (Wang et al., 2017). These datasets cover agriculture, remote sensing, and medical data, with substantial domain discrepancies.

In implementation, we set the similarity threshold to 0.3. We adopt ViT-S as our backbone network and initialize it with DINO pretraining on ImageNet following (Caron et al., 2021; Fu et al., 2023; Zhang et al., 2022). Additionally, our

*Table 2.* Ablation study by 5-shot.

| Method | CropDisease | EuroSAT | ISIC2018 | ChestX | Ave. |
|---|---|---|---|---|---|
| Baseline | $94.29_{\pm0.17}$ | $89.43_{\pm0.17}$ | $45.83_{\pm0.23}$ | $26.07_{\pm0.17}$ | 63.91 |
| + Warm up disruption | $95.31_{\pm0.15}$ | $89.61_{\pm0.17}$ | $48.83_{\pm0.23}$ | $27.01_{\pm0.17}$ | 65.19 |
| + Balanced disruption | $\mathbf{96.02}_{\pm0.14}$ | $\mathbf{90.42}_{\pm0.17}$ | $\mathbf{52.36}_{\pm0.23}$ | $\mathbf{28.23}_{\pm0.18}$ | **66.76** |
| (a) Remove Position Embedding | $95.02_{\pm0.16}$ | $89.96_{\pm0.16}$ | $47.40_{\pm0.24}$ | $26.85_{\pm0.17}$ | 64.81 |
| (b) Shuffled Patches | $95.03_{\pm0.15}$ | $88.67_{\pm0.18}$ | $48.04_{\pm0.23}$ | $27.08_{\pm0.17}$ | 64.70 |
| (c) Shuffle Patch Amp. | $94.75_{\pm0.16}$ | $88.78_{\pm0.17}$ | $49.67_{\pm0.23}$ | $27.04_{\pm0.17}$ | 65.06 |
| (d) Shuffle Patch Phase | $94.94_{\pm0.16}$ | $88.56_{\pm0.18}$ | $49.47_{\pm0.23}$ | $26.96_{\pm0.17}$ | 64.98 |
| (e) Shuffle Image Amp. | $94.73_{\pm0.16}$ | $88.96_{\pm0.17}$ | $47.95_{\pm0.23}$ | $26.79_{\pm0.17}$ | 64.61 |
| (f) Shuffle Cluster Amp. w/ Balancing | $95.28_{\pm0.15}$ | $89.13_{\pm0.17}$ | $49.51_{\pm0.23}$ | $27.85_{\pm0.17}$ | 65.44 |
| (g) Shuffle Patch Amp. w/ Balancing | $96.05_{\pm0.14}$ | $89.25_{\pm0.17}$ | $50.85_{\pm0.23}$ | $28.12_{\pm0.18}$ | 65.94 |

model leverages the Adam optimizer(Kingma & Ba, 2017) for 50 epochs, with a learning rate of $10^{-6}$ assigned to the backbone network and $10^{-3}$ to the classifier, respectively. Experiments are conducted on NVIDIA GeForce RTX 3090 GPUs.

## 4.2. Comparison with State-of-the-Art Works

We report our comparison with state-of-the-art works for the 1-shot and 5-shot configurations in Tab.1, respectively. To ensure a fair assessment, we separately compare works that have undergone finetuning (FT) and those that have not. The asterisk (*) indicates a transductive setting. Notably, our results demonstrate superior average performance across all configurations and consistently surpass existing works.

## 4.3. Ablation Study

The ablation study for each module is presented in Tab.2. It is evident that all types of our proposed methods aimed at disrupting the continuity of image patches contribute positively to the performance in the target domains. Additionally, we conduct a comparative analysis between our approaches and other similar works to substantiate the rationale underlying our design choices.

### 4.3.1. VERIFICATION OF BREAKING THE CONTINUITY

To study the contribution of our methods, we compare our four types of methods for disrupting the continuity of image patches with the baseline model in Tab. 2a. It is evident that all four methods we propose outperform the baseline model significantly on the target domain, thereby demonstrating the effectiveness of our approaches.

### 4.3.2. SHUFFLING AMPLITUDE IN PATCH DIMENSION

We study the impact of shuffling amplitude in patch dimension and directly contrast it with shuffling amplitude across entire images (e), an approach that has been extensively explored in many previous works (as shown in Table 2b). Our findings reveal that the performance of the latter is inferior to ours, suggesting that manipulating image patches offers a more effective means of enhancing the transferability of ViT-based models. Moreover, by gradually reducing the size of maintained continuity to clusters (f) or patches (g), we can see the performance consistently increases, verifying

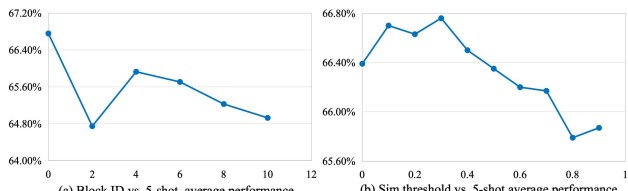

*Figure 6.* (a) Applying our approach to any layer results in performance enhancements, but the greatest improvement is achieved when it is applied to the input layer. (b) A relatively small similarity threshold can more effectively balance the style.

the importance of disrupting the continuity.

### 4.3.3. BALANCED DISRUPTION AND WARMING UP

Comparing the default shuffling of amplitude (c) and the balanced shuffling (g), we can see the balancing operation consistently increases the performance. Moreover, comparing the disruption without warming up (g) and our final performance, we can see the warming up step can also consistently improve the performance.

## 4.4. Sensitivity Study of Hyper-parameters

We conduct an analysis of the hyper-parameters presented in Fig.6 and 7, and see that:

(1) The input layer demonstrates effectiveness. As illustrated in Fig. 6a, merely applying our approach within the first block (i.e., the input layer) results in a substantial improvement in the model performance in target domains. Moreover, the application of our method to any layer yields notable improvements compared to not using it, further confirming the validity of our approach.

(2) A moderately small similarity distance threshold for dividing patches into clusters yields optimal results but should not be too minute, whereas a smaller threshold refers to larger clusters. As depicted in Fig. 6b, reducing it initially leads to a steady improvement in performance in the target domains. However, once this threshold is below approximately 30%, performance begins to decline.

(3) A larger standard deviation of sampling the proportions $p$ in Eq. 21 yields optimal results. As illustrated in Fig. 7a, increasing the standard deviation leads to a steady enhancement in performance until it reaches a plateau. This suggests that a larger standard deviation results in greater variability among patches and a higher degree of disrupted continuity, which is more beneficial for generalization in target domains.

## 4.5. Verification of model generalization

### 4.5.1. QUANTITATIVE STUDY

As depicted in Fig. 7b, we assess the domain similarity of the features extracted from the trained backbone network

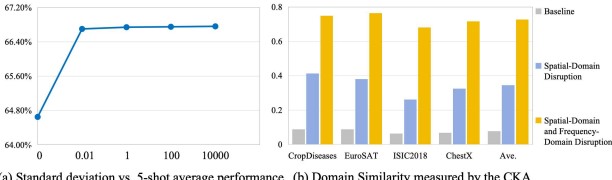

(a) Standard deviation vs. 5-shot average performance    (b) Domain Similarity measured by the CKA

*Figure 7.* (a) A larger standard deviation indicates greater discrepancy and discontinuity among image patches, and consequently, the performance on the target domain improves accordingly. This demonstrates that disrupting continuity is effective in reducing domain discrepancies. (b) Employing our method markedly boosts the similarity across domains.

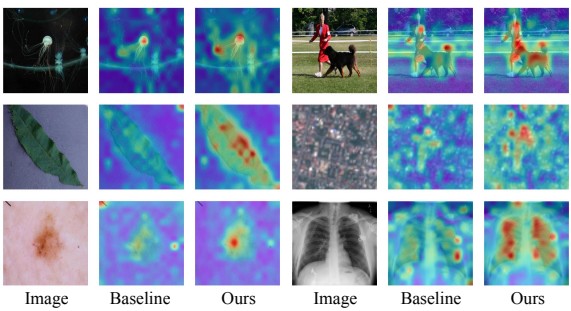

Image    Baseline    Ours    Image    Baseline    Ours

*Figure 8.* The heatmap for the source domain displayed in the first row illustrates that our method takes into account smaller patterns scattered throughout the image, collectively forming a broader perceptual area. The heatmaps for the target domains shown in the following two rows demonstrate that our approach effectively enhances ViT's perception of the target domain.

between the source and target domains using the CKA similarity metric. The results reveal that our model substantially elevates the domain similarity, suggesting that our model acquires domain-agnostic information.

### 4.5.2. QUALITATIVE STUDY

The visualization of the attention maps in both the source and target domains is presented in Fig. 8. In contrast to the baseline, which exhibits dispersed attention, our model not only focuses on the most discriminative large patterns but also attends to smaller, less discriminative patterns scattered across the image, forming a larger perceptual field. By transferring these patterns to the target domain, our model achieves a better perception of the target domain.

## 5. Related Work

**Cross-Domain Few-Shot Learning(CDFSL)** was introduced in FWT(Tseng et al., 2020), and a novel benchmark for this field has been established in BSCD-FSL(Guo et al., 2020). It strives to transfer knowledge from a proficiently trained source domain to a separate target domain characterized by limited labeled data and a large domain gap.

This field is primarily explored through two approaches: transferring-based methods (Zhou et al., 2023; Zou et al., 2024a), which adapt pre-trained models from extensive source datasets to target domains with scant data, and meta-learning approaches(Fu et al., 2022; Hu & Ma, 2022), which emphasize training models to swiftly adapt to novel tasks. However, no studies have explored the impact of continuity of ViT on cross-domain scenarios.

**Continuity**. ViT was introduced by (Dosovitskiy et al., 2021), which adopts an input approach by dividing images into patches and explores that when the input images are larger, the model is able to capture more detailed information. Further investigation is conducted by them on the incorporation of positional encoding, revealing that absolute positional encoding causes ViT to lack the translation invariance required for image processing. Then, (Liu et al., 2021) proposes a relative positional encoding to preserve invariance but is highly demanding of computational resources. Alternatively, (Chu et al., 2021) proposes a conditional positional encoding to address ViT's translation invariance while conserving computational resources. (Li et al., 2019; Wertheimer et al., 2021; Rong et al., 2023) find that the low-level local visual features can be more easily transferred to the target domain than those high-level semantic features. However, none of these studies specifically explore the impact of continuity on cross-domain performance. We are the first to investigate the influence of continuity on cross-domain performance.

## 6. Conclusion

In this paper, we find an intriguing phenomenon that breaking image tokens' continuity affects differently on the performance of source and target domains. We delve into this phenomenon for an interpretation, which inspires us to further propose a method to break the continuity, encouraging the model to rely less on large patterns and more on small patterns for recognition. Extensive experiments on four CDFSL benchmarks validate our rationale and effectiveness.

## Acknowledgments

This work is supported by the National Natural Science Foundation of China under grants 62206102; the National Key Research and Development Program of China under grant 2024YFC3307900; the National Natural Science Foundation of China under grants 62436003, 62376103 and 62302184; Major Science and Technology Project of Hubei Province under grant 2024BAA008; Hubei Science and Technology Talent Service Project under grant 2024DJC078; and Ant Group through CCF-Ant Research Fund. The computation is completed in the HPC Platform of Huazhong University of Science and Technology.

## Impact Statement

We propose a CD-FSL method that disrupts the continuity of image tokens within the Vision Transformer (ViT) architecture. This strategy encourages the model to reduce its dependence on large patterns and instead rely more on smaller, highly transferable patterns to the target domain. Additionally, our approach holds promise for application in other fields, including domain generalization, domain adaptation, and few-shot class-incremental learning, where enhancing model transferability poses a universal challenge. Although our evaluations concentrate on four distinct target domains, it is important to note that these may not exhaustively cover all potential real-world scenarios. Hence, further evaluation across a broader spectrum of target domains is crucial to validate our approach in more realistic and diverse settings.

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

# Appendix for Revisiting Continuity of Image Tokens for Cross-Domain Few-shot Learning

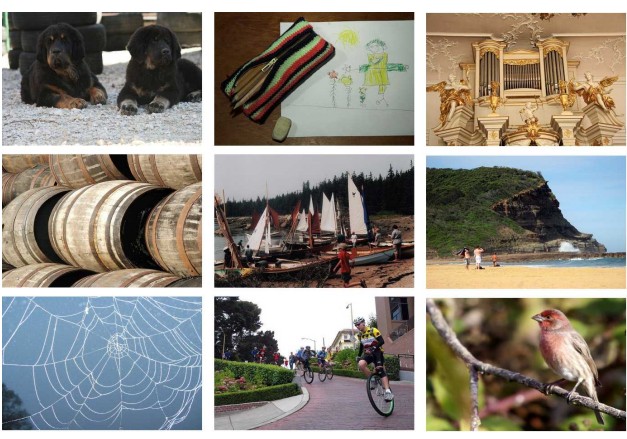

*Figure 9.* Samples of the *mini*ImageNet dataset.

## A. Dataset Description

***mini*ImageNet** (Vinyals et al., 2016) is a meticulously selected subset derived from the extensive ImageNet dataset (Deng et al., 2009). It encompasses 100 categories, with each category represented by 600 natural images, totaling 60,000 images. Consistent with recent research (Zou et al., a;b), we utilize the training portion of *mini*ImageNet as our source domain dataset. This comprises 64 classes and a total of 38,400 images, with representative samples displayed in Fig. 9.

Furthermore, as illustrated in Fig. 10, we adopt the methodology outlined in (Guo et al., 2020) and employ datasets from four distinct domains as our target domains. These domains include plant disease images, surface satellite imagery, skin disease images, and chest X-ray images, which will be elaborated on in the following sections.

**CropDiseases** (Mohanty et al., 2016) is specifically designed for the recognition of agricultural diseases. It contains 43,456 images across 38 different classes, featuring a diverse range of crops, both healthy and diseased, each labeled with a specific disease category.

**EuroSAT** (Helber et al., 2019) is a comprehensive dataset comprising an extensive collection of satellite imagery of Earth's surface. It encompasses a total of 27,000 images, meticulously categorized into 10 diverse classes, spanning a broad spectrum of geographical and topographical attributes.

**ISIC2018** (Codella et al., 2019) is a specialized dataset

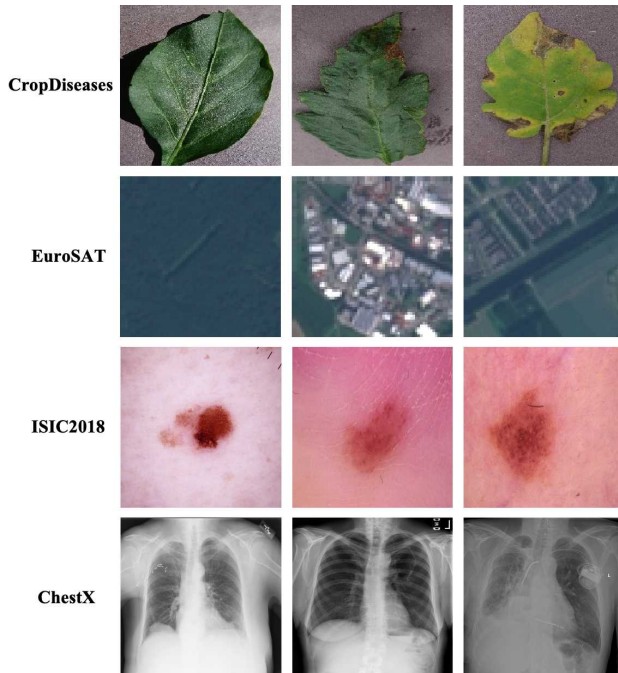

*Figure 10.* Samples of the CropDiseases, EuroSAT, ISIC2018 and ChestX datasets.

dedicated to medical imaging, particularly focusing on the classification of skin lesions. This dataset boasts 10,015 images distributed across 7 distinct categories, serving as a pivotal resource for diagnosing skin diseases.

**ChestX** (Wang et al., 2017) is another medical imaging dataset that centers on chest X-rays. It comprises 25,847 images, meticulously categorized into 7 unique classes, offering invaluable data for tasks related to chest disease classification.

## B. Centered Kernel Alignment

Centered Kernel Alignment (CKA) (Kornblith et al., 2019) is a statistical technique designed to quantify the similarity between representations learned by disparate neural networks. Originating from kernel methods, CKA excels in comparing high-dimensional representations. To compute CKA between two sets of data representations $X \in \mathbb{R}^{n \times d}$ and $Y \in \mathbb{R}^{n \times d}$, we first calculate the Gram matrices $K = XX^\top$ and $L = YY^\top$. These matrices encapsulate the inner products between all pairs of data points within

*Table 3.* Comparison with more state-of-the-art works based by 5-way 1-shot accuracy.

| Method | Backbone | FT | Mark | ChestX | ISIC2018 | EuroSAT | CropDiseases | Average |
|---|---|---|---|---|---|---|---|---|
| GNN + FT (Tseng et al., 2020) | ResNet10 | × | ICLR-20 | 22.00 | 30.22 | 55.53 | 60.74 | 42.12 |
| MN + AFA (Hu & Ma, 2022) | ResNet10 | × | ECCV-22 | 22.11 | 32.32 | 61.28 | 60.71 | 44.10 |
| GNN + ATA (Wang & Deng, 2021) | ResNet10 | × | IJCAI-21 | 22.10 | 33.21 | 61.35 | 67.47 | 46.53 |
| GNN + AFA (Hu & Ma, 2022) | ResNet10 | × | ECCV-22 | 22.92 | 33.21 | 63.12 | 67.61 | 46.97 |
| LDP-net (Zhou et al., 2023) | ResNet10 | × | CVPR-23 | 23.01 | 33.97 | 65.11 | 69.64 | 47.18 |
| FLoR (Zou et al., 2024a) | ResNet10 | × | CVPR-24 | 23.11 | **38.11** | 62.90 | 73.64 | 49.69 |
| MEM-FS (Walsh et al., 2023) | ViT-S | × | TIP-23 | 22.76 | 32.97 | 68.11 | 81.11 | 51.24 |
| StyleAdv (Fu et al., 2023) | ViT-S | × | CVPR-23 | 22.92 | 33.05 | 72.15 | 81.22 | 52.34 |
| FLoR (Zou et al., 2024a) | ViT-S | × | CVPR-24 | 22.78 | 34.20 | 72.39 | 81.81 | 52.80 |
| DAMIM (Ma et al., 2024) | ViT-S | × | AAAI-25 | 22.97 | 34.66 | 72.87 | 82.34 | 53.21 |
| CD-CLS (Zou et al., b) | ViT-S | × | NeurIPS-24 | 22.93 | 34.21 | 74.08 | 83.51 | 53.68 |
| AttnTemp (Zou et al., a) | ViT-S | × | NeurIPS-24 | 23.19 | 34.92 | 74.35 | 84.02 | 54.12 |
| **ReCIT** | ViT-S | × | **Ours** | **23.27** | 35.13 | **74.56** | **84.76** | **54.43** |
| PMF (Shell Xu, 2022) | ViT-S | ✓ | CVPR-22 | 21.73 | 30.36 | 70.74 | 80.79 | 50.91 |
| FLoR (Zou et al., 2024a) | ViT-S | ✓ | CVPR-24 | 23.26 | 35.49 | 73.09 | 83.55 | 53.85 |
| StyleAdv (Fu et al., 2023) | ViT-S | ✓ | CVPR-23 | 22.92 | 33.99 | 74.93 | 84.11 | 53.99 |
| DAMIM (Ma et al., 2024) | ViT-S | ✓ | AAAI-25 | 23.38 | 36.35 | 73.61 | 83.90 | 54.31 |
| CD-CLS (Zou et al., b) | ViT-S | ✓ | NeurIPS-24 | 23.39 | 35.56 | 74.97 | 84.54 | 54.62 |
| AttnTemp (Zou et al., a) | ViT-S | ✓ | NeurIPS-24 | 23.63 | 38.05 | 75.09 | 84.78 | 55.39 |
| **ReCIT** | ViT-S | ✓ | **Ours** | **23.84** | **38.48** | **75.23** | **85.92** | **55.87** |
| LDP-net[*] (Zhou et al., 2023) | ResNet10 | ✓ | CVPR-23 | 22.21 | 33.44 | 73.25 | 81.24 | 52.54 |
| TPN + ATA[*] (Wang & Deng, 2021) | ResNet10 | ✓ | IJCAI-21 | 22.45 | 35.55 | 70.84 | 82.47 | 52.83 |
| RDC[*] (Li et al., 2022) | ResNet10 | ✓ | CVPR-22 | 22.32 | 36.28 | 70.51 | 85.79 | 53.73 |
| MEM-FS + RDA[*] (Walsh et al., 2023) | ViT-S | ✓ | TIP-23 | 23.85 | 37.07 | 75.91 | 83.74 | 55.14 |
| DAMIM[*] (Ma et al., 2024) | ViT-S | ✓ | AAAI-25 | 23.91 | 38.07 | 77.23 | 86.74 | 56.49 |
| CD-CLS (Zou et al., b) | ViT-S | ✓ | NeurIPS-24 | 23.88 | 37.20 | 78.41 | 87.39 | 56.72 |
| AttnTemp (Zou et al., a) | ViT-S | ✓ | NeurIPS-24 | 23.96 | **40.13** | 77.40 | 87.58 | 57.23 |
| **ReCIT**[*] | ViT-S | ✓ | **Ours** | **24.42** | 39.39 | **78.84** | **88.45** | **57.78** |

their respective feature spaces. Subsequently, we proceed to center the Gram matrices by applying the following process:

$$K_d = HKH, \quad (22)$$

$$L_d = HLH, \quad (23)$$

where $H = I_n - \frac{1}{n}1_n1_n^\top$ is the centering matrix, $I_n$ is the identity matrix, and $1_n$ is a vector of ones. Finally, the centered kernel alignment is computed as:

$$\mathrm{CKA}(K, L) = \frac{\mathrm{Tr}(K_d L_d)}{\sqrt{\mathrm{Tr}(K_d^2)}\sqrt{\mathrm{Tr}(L_d^2)}}, \quad (24)$$

where Tr denotes the trace of a matrix. CKA is an exceptionally valuable tool for assessing similarity, particularly when comparing different datasets or models. High CKA values serve as indicators of strong similarity, whereas low values point to notable differences. In this paper, we leverage CKA to evaluate the similarity of features across various datasets and models. Specifically, for each dataset, we extract features from the model and compute the CKA similarity between the source and target domains, aiming to gain insights into the model's generalization capabilities across diverse data distributions.

## C. Comparison with more SOTAs

As depicted in Tab. 3 and 4, we undertake a comprehensive comparison of diverse ViT-based and CNN-based methodologies in the context of CDFSL tasks. Our proposed methods consistently outperform all other approaches, attaining superior performance. These results underscore the effectiveness of our approach.

## D. Source Domain Performance

We have provided the performance on the source domain (*mini*ImageNet) for reference in Tab. 5.

Although there is a slight performance trade-off on the source domain, this aligns with our hypothesis: disrupting token continuity prioritizes learning smaller, transferable patterns over domain-specific holistic features. Crucially, the significant gains on target domains demonstrate the effectiveness of our approach for cross-domain adaptation.

## E. Detailed related work

**Cross-Domain Few-Shot Learning (CDFSL)** (Zhou et al., 2023) focuses on training a model on the source domain that can generalize well to target domain with limited ex-

*Table 4.* Comparison with more state-of-the-art works by 5-way 5-shot accuracy.

| Method | Backbone | FT | Mark | ChestX | ISIC2018 | EuroSAT | CropDiseases | Average |
|---|---|---|---|---|---|---|---|---|
| MN + AFA (Hu & Ma, 2022) | ResNet10 | × | ECCV-22 | 23.18 | 39.88 | 69.63 | 80.07 | 53.19 |
| GNN + FT (Tseng et al., 2020) | ResNet10 | × | ICLR-20 | 24.28 | 40.87 | 78.02 | 87.07 | 57.06 |
| GNN + ATA (Wang & Deng, 2021) | ResNet10 | × | IJCAI-21 | 24.32 | 44.91 | 83.75 | 90.59 | 60.39 |
| LDP-net (Zhou et al., 2023) | ResNet10 | × | CVPR-23 | 26.67 | 48.06 | 82.01 | 89.40 | 61.29 |
| GNN + AFA (Hu & Ma, 2022) | ResNet10 | × | ECCV-22 | 25.02 | 46.01 | 85.58 | 88.06 | 61.67 |
| FLoR (Zou et al., 2024a) | ResNet10 | × | CVPR-24 | 26.70 | 51.44 | 80.87 | 91.25 | 62.32 |
| MEM-FS (Walsh et al., 2023) | ViT-S | × | TIP-23 | 26.67 | 47.38 | 86.49 | 93.74 | 63.57 |
| StyleAdv (Fu et al., 2023) | ViT-S | × | CVPR-23 | 26.97 | 47.73 | 88.57 | 94.85 | 64.53 |
| FLoR (Zou et al., 2024a) | ViT-S | × | CVPR-24 | 26.71 | 49.52 | 90.41 | 95.28 | 65.48 |
| DAMIM (Ma et al., 2024) | ViT-S | × | AAAI-25 | 27.28 | 50.76 | 89.50 | 95.52 | 65.77 |
| CD-CLS (Zou et al., b) | ViT-S | × | NeurIPS-24 | 27.23 | 50.46 | **91.04** | 95.68 | 66.10 |
| AttnTemp (Zou et al., a) | ViT-S | × | NeurIPS-24 | 27.72 | **53.09** | 90.13 | 95.53 | 66.62 |
| **ReCIT** | ViT-S | × | **Ours** | **28.23** | 52.36 | 90.42 | **96.02** | **66.76** |
| PMF (Shell Xu, 2022) | ViT-S | ✓ | CVPR-22 | 27.27 | 50.12 | 85.98 | 92.96 | 64.08 |
| StyleAdv (Fu et al., 2023) | ViT-S | ✓ | CVPR-23 | 26.97 | 51.23 | 90.12 | 95.99 | 66.08 |
| FLoR (Zou et al., 2024a) | ViT-S | ✓ | CVPR-24 | 27.02 | 53.06 | 90.75 | 96.47 | 66.83 |
| DAMIM (Ma et al., 2024) | ViT-S | ✓ | AAAI-25 | 27.82 | 54.86 | 91.18 | 96.34 | 67.55 |
| CD-CLS (Zou et al., b) | ViT-S | ✓ | NeurIPS-24 | 27.66 | 54.69 | 91.53 | 96.27 | 67.54 |
| AttnTemp (Zou et al., a) | ViT-S | ✓ | NeurIPS-24 | 28.03 | 54.91 | 90.82 | 96.66 | 67.61 |
| **ReCIT** | ViT-S | ✓ | **Ours** | **28.88** | **54.91** | **91.58** | **96.85** | **68.06** |
| ConFeSS[*] (Das et al., 2022) | ResNet10 | ✓ | ICLR-2022 | 27.09 | 48.85 | 84.65 | 88.88 | 62.37 |
| LDP-net[*] (Zhou et al., 2023) | ResNet10 | ✓ | CVPR-23 | 26.88 | 48.44 | 84.05 | 91.89 | 62.82 |
| RDC[*] (Li et al., 2022) | ResNet10 | ✓ | CVPR-22 | 25.07 | 49.91 | 84.29 | 93.30 | 63.14 |
| TPN + ATA[*] (Wang & Deng, 2021) | ResNet10 | ✓ | IJCAI-21 | 24.74 | 49.83 | 85.47 | 93.56 | 63.40 |
| MEM-FS + RDA[*] (Walsh et al., 2023) | ViT-S | ✓ | TIP-23 | 27.98 | 51.02 | 88.77 | 95.04 | 65.70 |
| DAMIM[*] (Ma et al., 2024) | ViT-S | ✓ | AAAI-25 | 28.10 | 55.44 | 91.08 | 96.49 | 67.78 |
| CD-CLS[*] (Zou et al., b) | ViT-S | ✓ | NeurIPS-24 | 28.25 | **55.66** | 91.68 | 96.62 | 68.05 |
| AttnTemp[*] (Zou et al., a) | ViT-S | ✓ | NeurIPS-24 | 28.41 | 55.22 | 91.34 | 96.74 | 67.93 |
| **ReCIT[*]** | ViT-S | ✓ | **Ours** | **28.97** | 55.60 | **91.72** | **96.97** | **68.32** |

*Table 5.* Ablation study of source-domain training by 5-shot.

| Method | Source Domain | Target Domain |
|---|---|---|
| Baseline | 97.78 | 63.91 |
| Ours | 96.33 | 66.76 |

CD-FSL by spanning distributions of source styles. IM-DCL (Xu et al., 2024) sets the entire feature as positive and negative sets to learn the query set without accessing the source domain. However, no studies have explored the impact of the continuity of ViT in cross-domain scenarios.

amples. Current methods can be grouped into two types: meta-learning-based approaches and transfer-learning-based ones. Meta-learning-based approaches (Zhang et al., 2018) aim at learning task-agnostic knowledge to learn new tasks efficiently, differing in their way of learning the parameter of the initial model on the base class data. MAML (Raghu et al., 2019) aims at learning an initial parameter that can quickly adapt to new tasks, while FWT (Tseng et al., 2020) uses a feature-wise transformation to learn representations with improved ability to generalization. An alternative way to tackle the problem is transfer-learning-based approaches, tackling the problem based on reusing the model trained on the base class data in a standard supervised learning way. Among these approaches, LRP (Sun et al., 2021) aims to use the explanation results to guide the learning process. STARTUP (Phoo & Hariharan, 2021), and Meta-FDMixup (Fu et al., 2021) mainly aim at defining relaxed settings for CD-FSL. Wave-SAN (Fu et al., 2022) tackles

