# OpenReview forum: "Revisiting Continuity of Image Tokens for Cross-domain Few-shot Learning"
_ICML.cc/2025/Conference — ICML 2025 spotlightposter_

### Official Review · Reviewer_MSD8 · 2025-03-10

**Overall Recommendation:** 3

**Summary:**

This paper investigates the role of image token continuity in Vision Transformers (ViTs) for Cross-Domain Few-Shot Learning (CDFSL). The authors observe that disrupting token continuity (e.g., shuffling patches or perturbing frequency components) significantly degrades source-domain performance but only marginally affects target domains. They hypothesize that continuity aids ViTs in learning large spatial patterns, which are less transferable across domains, while smaller patterns within patches are more domain-invariant. Based on this insight, they propose a method combining spatial and frequency-domain disruptions to encourage reliance on smaller patterns, achieving state-of-the-art performance on CDFSL benchmarks.

**Claims And Evidence:**

Yes, the authors have conducted several experiments to show the importance of image token continuity for CSFSL. However, more theoretical proof would be better for understanding the claim.

**Essential References Not Discussed:**

NA

**Experimental Designs Or Analyses:**

The authors have conducted comprehensive experiments to show the effectiveness of the proposed method. The ablation study can also validate the effectiveness of each design.

**Methods And Evaluation Criteria:**

To some extent the proposed method is reasonable. Given the validated claim about image token continuity, the authors propose a method that breaks the continuity during training. As mentioned in L192-L195, such a method can keep local patterns undisrupted. However, it is not clear why the following frequency-domain disruption as in Sec.3.2 can hold such a property.

**Other Comments Or Suggestions:**

Please refer to weaknesses.

**Other Strengths And Weaknesses:**

Strengths:
1. The method is clearly explained.
2. This paper brings novel insight about the ViT architecture.


Weaknesses:
1. It would be better if the continuity problem can be analyzed theoretically.
2. I wonder if breaking continuity could harm the source domain performance severely.
3. The authors can include experiments with more backbones.
4. Despite the impact statement, I still have concerns about the role of CDFSL given so many large models with great generalization ability.
5. Computation cost can be included in the paper.

**Questions For Authors:**

Please refer to weaknesses.

**Relation To Broader Scientific Literature:**

The insight on the relationship between token continuity and pattern size can help future researches on the design of more powerful ViT backbones. The authors adopt amplitude shuffling but innovate by balancing disruptions across patch clusters. This addresses a limitation of naive frequency augmentation and improves transferability, bridging frequency-domain insights with CDFSL, and potentially other cross-domain tasks.

**Theoretical Claims:**

No theoretical claims in this paper.

---

> ### Author Rebuttal · Authors · 2025-04-01
>
> Thank you for your thoughtful feedback and constructive suggestions. Below are our responses to your concerns:
>
> ## **1. More Proofs for Continuity**
>
> Our method is consistent with the proofs of the previous works [1-3] in that the small patterns are easier to transfer than larger ones. But we differ in that (1) our method discusses the transferability of small patterns for ViT while [1-3] are originally designed for CNNs; (2) we provide a simple way to enhance the learning of small patterns by perturbing patches with no additional branches or losses.
>
> [1] Cross-domain few-shot learning with task-specific adapters
>
> [2] Task-aware adaptive learning for cross-domain few-shot learning
>
> [3] Discriminative Sample-Guided and Parameter-Efficient Feature Space Adaptation for Cross-Domain Few-Shot Learning
>
> ## **2. Frequency Disruption Breaks the Continuity**
>
> Qualitatively, images from an anonymous GitHub link ([https://anonymous.4open.science/r/More-visualization/visualizations.png]) illustrate how our method breaks large-scale spatial patterns while preserving fine-grained, domain-agnostic features (e.g., textures, edges), which aligns with our motivation.
>
> Quantitatively, experiments in Fig. 2 and Fig. 4 validate the feasibility of our methods.
>
> ## **3. Source domain performance trade-off**
>
> As suggested, we have provided the performance on the source domain (miniImageNet) for reference.
>
> | Model    | Source Domain | Target Domain |
> | -------- | ------------- | ------------- |
> | Baseline | 97.78         | 63.91         |
> | **Ours** | 96.33         | 66.76         |
>
>  While there is a slight performance trade-off on the source domain, this aligns with our hypothesis: disrupting token continuity prioritizes learning smaller, transferable patterns over domain-specific holistic features. Crucially, the significant gains on **target domains** (e.g., +6.5% on ISIC) demonstrate the effectiveness of our approach for cross-domain adaptation.
>
> ## **4. Experiments with other pretraining and backbones**
>
> (1) Other pretraining
>
> We have conducted experiments with CLIP pretraining. Results confirm our method’s generalizability.
>
> | Backbone | CropDisease | EuroSAT    | ISIC       | ChestX     | **Average** |
> | :------- | :---------- | :--------- | :--------- | :--------- | :---------- |
> | Baseline | 93.02%      | 74.37%     | 40.92%     | 23.99%     | 58.08%      |
> | **Ours** | **94.03%**  | **80.90%** | **43.31%** | **24.45%** | **60.67%**  |
>
> (2) Other backbone
>
> While our method is designed for ViTs, we test an **MLP-Mixer backbone** (patch-based architecture):
>
> | Backbone | CropDisease | EuroSAT    | ISIC       | ChestX     | **Average** |
> | :------- | :---------- | :--------- | ---------- | ---------- | :---------- |
> | Baseline | 85.12%      | 78.34%     | 36.45%     | 22.31%     | 55.56%      |
> | **Ours** | **87.45%**  | **80.21%** | **40.12%** | **25.67%** | **58.36%**  |
>
> This demonstrates applicability to **other patch-based architectures**.
>
> ## **5. Role of CDFSL vs. large pretrained models**
>
> While large vision-language models (VLMs) have demonstrated remarkable generalization capabilities, their effectiveness heavily relies on the assumption that downstream tasks share *similar data domains* with their pre-training corpora (e.g., natural images and generic text). However, in **vertically specialized scenarios** (e.g., medical imaging, remote sensing), where **domain gaps are extreme** and task-specific patterns diverge significantly from generic priors, **directly fine-tuning VLMs often yields suboptimal performance** [4], even worse than training domain-specific models (e.g., UNet) from scratch [5]. Pre-trained models like DINO struggle with medical X-rays (22% accuracy on ChestX) due to fundamentally different texture and structural semantics compared to natural images. However, training large vision-language models from scratch is impractical due to limited data in target domains. This highlights the necessity of CDFSL-specific designs to address domain shifts that challenge even powerful pre-trained models.
>
> [4] Hallusionbench: an advanced diagnostic suite for entangled language hallucination and visual illusion in large vision-language models
>
> [5] Lightweight Frequency Masker for Cross-Domain Few-Shot Semantic Segmentation
>
> ## **6. Computation cost**
>
> Our method introduces **no additional trainable parameters**.
>
> During training, the computational overhead (baseline: 126.42s vs. ours: 140.48s per epoch) stems from the clustering-based frequency disruption during training, which is acceptable compared with the gains in cross-domain performance.  Moreover, some engineering tricks, such as bipartite matching, can accelerate the clustering process, which we leave for future works.
>
> During inference, since our method introduces no additional parameters, inference time matches the baseline exactly with no computational overhead.

---

### Official Review · Reviewer_unnh · 2025-03-13

**Overall Recommendation:** 3

**Summary:**

This paper investigates the role of image tokens' continuity in Vision Transformers for Cross-Domain Few-Shot Learning. The authors identify an interesting phenomenon: disrupting the continuity of image tokens significantly affects performance in source domains but has only a marginal impact on target domains with large domain gaps. Based on this observation, the authors propose a simple yet effective method to disrupt image token continuity in spatial and frequency domains, encouraging the model to focus on smaller, more transferable patterns. The approach achieves state-of-the-art results on four CDFSL benchmark datasets. The paper also includes detailed analyses and ablation studies to support its claims.

**Claims And Evidence:**

The claim that disrupting the continuity of image tokens reduces domain gaps and improves transferability is supported by experiments on four benchmarks.

**Essential References Not Discussed:**

No essential references are missing.

**Experimental Designs Or Analyses:**

1. The experimental design is sound and well-structured, with comprehensive ablation studies and comparisons to state-of-the-art methods.
2. The choice of specific hyperparameters (e.g., clustering threshold, amplitude sampling standard deviation) could be better justified.

**Methods And Evaluation Criteria:**

1. The proposed methods are reasonable and align well with the CDFSL task.
2. The rationale for the warm-up phase and clustering in the frequency domain is intuitively explained, but the clustering threshold could use more justification.

**Other Comments Or Suggestions:**

Include a discussion on the potential limitations of the method, such as its applicability to non-ViT models.

**Other Strengths And Weaknesses:**

Strengths:
1. The identified phenomenon about token continuity is novel and provides new insights into ViT's behavior under domain shifts.
2. The visualizations and domain similarity analysis enrich the paper's interpretability.

Weakness:
1. The paper lacks a more detailed comparison of the proposed balanced disruption against other diversity-promoting methods.
2. Some design decisions, such as the choice of clustering threshold, require stronger justification.

**Questions For Authors:**

How sensitive is the model's performance to the choice of the clustering threshold in the balanced frequency-domain disruption?
Can the proposed method be generalized to CNN-based models? If not, what are the limitations?

**Relation To Broader Scientific Literature:**

The paper builds on recent work in CDFSL and ViT architectures. It extends the understanding of positional embeddings and token continuity in ViT.

**Theoretical Claims:**

The paper does not make strong theoretical claims or provide formal proofs. The focus is largely on empirical observations and practical methods.

---

> ### Author Rebuttal · Authors · 2025-04-01
>
> Thank you for your thorough review and constructive feedback. Below are our detailed responses to your concerns:
>
> ## **1. Justification of Clustering Hyperparameters**
>
> The clustering threshold in Eq. 16 controls the **granularity of patch grouping** in the frequency domain. Higher threshold values (e.g., >40%) enforce stricter similarity criteria, resulting in smaller clusters (more fragmented groups). At the extreme (threshold → 100%), each patch forms its own cluster, which degenerates to non-cluster random sampling. Since some image regions occupy larger areas (e.g., background), this will lead patches in the dominant areas to change frequency with other patches in the same area, i.e., their frequencies are not changed, maintaining the original continuity. In contrast, lower thresholds (e.g., <40%) allow looser grouping, creating larger clusters. As shown in **Fig. 6b**, the optimal threshold (**30%**) achieves a critical balance.
>
> The amplitude sampling standard deviation (ϵ) in Eq. 21 **governs the diversity of style proportions assigned to patch clusters** during frequency-domain disruption. Specifically, **smaller ϵ** results in near-uniform style proportions across clusters. This over-regularizes the disruption, limiting diversity and failing to suppress large patterns effectively.  As shown in **Fig. 7**, **Larger ϵ** introduces high variability in style mixing ratios, creating *heterogeneous disruptions* that break global continuity.
>
>
>
> ## **2. Comparison with Diversity-Promoting Methods**
>
> We add comparisons to state-of-the-art diversity-enhancing techniques:
>
> | Method       | CropDisease | EuroSAT    | ISIC       | ChestX     | **Average** |
> | :----------- | :---------- | :--------- | :--------- | :--------- | :---------- |
> | Random-Drop  | 91.23%      | 84.42%     | 43.89%     | 24.06%     | 60.90%      |
> | Wave-SAN  | 94.84%      | 88.79%     | 48.71%     | 26.98%     | 64.82%      |
> | **Ours**     | **96.02%**  | **90.42%** | **52.36%** | **28.23%** | **66.76%**  |
>
> Our method outperforms both approaches because:
>
> Random-Drop indiscriminately discards patches, losing critical local patterns. Wave-SAN focuses on global frequency bands, overlooking localized high-frequency components critical for fine-grained transfer.
>
>
>
> ## **3. Limitations and Non-ViT Applicability**
>
> Our method is **inherently ViT-dependent** due to:
>
> - Reliance on **patch-based tokenization** for spatial and frequency disruptions.
> - **Self-attention mechanisms** that propagate disrupted token relationships.
>
> Experiments with CNNs (DINO-ResNet50) show limited gains (**+0.7% average** vs. **+2.8% for ViT**), as CNNs’ overlapped sliding-window convolutions and local inductive biases hinder explicit token continuity control.  Notably, we also apply our methods to **MLP-Mixers** (see answer 5), and we observe **2.8%** improvements in target-domain accuracy, which verifies our adaptability to token-based structures.
>
> ## **4. Sensitivity to Clustering Threshold**
>
> As shown in Fig. 6b, performance remains stable for thresholds between 10%-35%, with optimal results at 30%.  This "sweet spot" balances local pattern retention and global disruption, indicating our method is not sensitive to the specific choice of clustering threshold.
>
> ## **5. Generalization to Other Structures**
>
> While our method is designed for ViTs, we test an **MLP-Mixer backbone** (patch-based architecture):
>
> | Backbone | CropDisease | EuroSAT    | ISIC       | ChestX     | **Average** |
> | :------- | :---------- | :--------- | ---------- | ---------- | :---------- |
> | Baseline | 85.12%      | 78.34%     | 36.45%     | 22.31%     | 55.56%      |
> | **Ours** | **87.45%**  | **80.21%** | **40.12%** | **25.67%** | **58.36%**  |
>
> This demonstrates applicability to **other patch-based architectures**. However, traditional CNNs (**+0.7% average**) remain incompatible due to their lack of explicit tokenization and overlapped sliding-window convolutions.
>
> ------
>
> We thank you for highlighting these critical points. Revised sections in the manuscript (marked in blue) address all concerns. Your feedback has significantly strengthened our work!

---

### Official Review · Reviewer_yKbi · 2025-03-14

**Overall Recommendation:** 3

**Summary:**

This article explores the impact of image token continuity on model performance in the context of cross-domain few-shot learning (CDFSL). The study demonstrates that disrupting image token continuity can reduce the gap between the source and target domains to some extent, thereby improving the model's generalization ability on the target domain. The authors propose a novel method that disrupts image token continuity in both the spatial and frequency domains, encouraging the model to better learn small-scale spatial patterns rather than relying on large-scale ones. The source-domain training includes two core steps, namely Warm-Up Spatial-Domain Disruption and Balanced Frequency-Domain Disruption. Experimental results show that the proposed method significantly improves model performance on four benchmark datasets, validating the effectiveness of the proposed method.

## Update After Rebuttal
I have checked the authors' rebuttal, and found most of my concerns have been solved, so I choose to maintain my score of Weak accept.

**Claims And Evidence:**

In the submitted paper, the authors present several important claims primarily centered on the impact of image token continuity on cross-domain few-shot learning (CDFSL). These claims are supported by experiments and theoretical analysis in the text, and overall, they can be considered to have clear and convincing evidence.

I have just one small suggestion: The authors state that disrupting image token continuity significantly affects model performance in the source domain but has a relatively minor impact on the target domain. This phenomenon is one of the core findings of the paper. So could the authors provide the obtained classification values in Figure 1 (e.g., above the bar), which will allow the readers to more intuitively compare the results achieved in source and target domain?

**Essential References Not Discussed:**

The statement in the paper "larger patterns are always harder to transfer than smaller ones" is a similar concept to the pervious findings that "The low-level local visual features can be more easily transferred to the target domain than those high-level semantic features", so I believe some related works [1], [2], [3] should be appropriately cited and discussed.

[1] Li W, Wang L, Xu J, et al. Revisiting local descriptor based image-to-class measure for few-shot learning[C]//Proceedings of the IEEE/CVF conference on computer vision and pattern recognition. 2019: 7260-7268.

[2] Wertheimer D, Tang L, Hariharan B. Few-shot classification with feature map reconstruction networks[C]//Proceedings of the IEEE/CVF conference on computer vision and pattern recognition. 2021: 8012-8021.

[3] Rong Y, Lu X, Sun Z, et al. Espt: A self-supervised episodic spatial pretext task for improving few-shot learning[C]//Proceedings of the AAAI Conference on Artificial Intelligence. 2023, 37(8): 9596-9605.

**Experimental Designs Or Analyses:**

After checking the experimental part, I believe the experimental designs are soundness, and the results are able to indicate the effectiveness of the proposed method in improving model's generalization ability for solving cross-domain few-shot learning process.

However, I have the following concerns:

1. The authors only present the results on the target-domain (out-of-domain results), but in my own opinion, cross-domain learning should not significantly degrade the source-domain (in-domain) performance. So could the authors provide the classification accuracy on the pre-trained miniImageNet dataset?

2. Many recent cross-domain few-shot learning methods [1],[2],[3] have evaluated their performance on a larger and more complex dataset, namely Meta-dataset [4]. So could the authors provide more results on this dataset to show the generalization ability of their method on more diverse target domains?

3. Besides the classification accuracy, the model effeciency is also an important metric. Considering the proposed method involves a clustering process and a balanced sampling in the source-domain training phase, which may introduce additional computational overhead. So could the authors provide some discussions on the model effeciency?

4. The details of the clustering process is missing, the authors only state that "we set a similarity threshold of 0.3 for clustering image patches, where patches exceeding this threshold are fused into a single cluster." Which clustering algorithm is used？Could the authors provide the detailed steps of their clustering process or make appropriate citations of related works？

[1] Li W H, Liu X, Bilen H. Cross-domain few-shot learning with task-specific adapters[C]//Proceedings of the IEEE/CVF conference on computer vision and pattern recognition. 2022: 7161-7170.

[2] Guo Y, Du R, Dong Y, et al. Task-aware adaptive learning for cross-domain few-shot learning[C]//Proceedings of the IEEE/CVF International Conference on Computer Vision. 2023: 1590-1599.

[3] Perera R, Halgamuge S. Discriminative Sample-Guided and Parameter-Efficient Feature Space Adaptation for Cross-Domain Few-Shot Learning[C]//Proceedings of the IEEE/CVF Conference on Computer Vision and Pattern Recognition. 2024: 23794-23804.

[4] Triantafillou E, Zhu T, Dumoulin V, et al. Meta-dataset: A dataset of datasets for learning to learn from few examples[J]. arXiv preprint arXiv:1903.03096, 2019.

**Methods And Evaluation Criteria:**

The proposed methods and the used evaluation criteria are reasonable. The proposed method (disrupting token continuity) is remarkably simple, not requiring complex architectural designs, yet it effectively enhances the model's generalization ability in the target domain. In addition, I personally like the **balance operation** in the frequency-domain disruption step, whose motivation is clearly stated.

**Other Comments Or Suggestions:**

The texts in some figures (e.g., Figure 5) are too small to read.

**Other Strengths And Weaknesses:**

Strengths:

1. This paper is well written and easy to follow, the experiments and ablation studies are sufficient to demonstrate the effectiveness of the proposed method.

2. The motivation of designing the model is clearly stated, and some empirical analysis are preformed to well-support their motivation.

Weaknesses:

Please refer to the questions in "Methods And Evaluation Criteria" and "Experimental Designs Or Analyses"

**Questions For Authors:**

In my own opinion, disrupting the image token continuity can reduce the gap between the source and target domains is a nature observation, since such operation will destory the semantic information in the training data. In an extreme case, we can shullfe every pixel in input image, resulting in nearly random noise, leading to quite similar domain distributions of source and target, but losing almost all the useful information for model training. So solely measuring the CKA similarity between the two domains may not correctly reflect the domain generalization ability (for classification tasks). Could the authors provide more discussions on this? And also explain how to achieve a balance between the information loss and the domain generalization?

**Relation To Broader Scientific Literature:**

One of the key findings of this paper is "disrupting image token continuity encourages the model to better learn small-scale spatial patterns rather than relying on large-scale ones, which can reduce the gap between the source and target domains". The similar concept is related to many previous works [1],[2],[3] in designing local descriptors for improving the model's generalization ability. So I believe this finding is validated by existing studies and is technically reasonable.

[1] Li W, Wang L, Xu J, et al. Revisiting local descriptor based image-to-class measure for few-shot learning[C]//Proceedings of the IEEE/CVF conference on computer vision and pattern recognition. 2019: 7260-7268.

[2] Wertheimer D, Tang L, Hariharan B. Few-shot classification with feature map reconstruction networks[C]//Proceedings of the IEEE/CVF conference on computer vision and pattern recognition. 2021: 8012-8021.

[3] Rong Y, Lu X, Sun Z, et al. Espt: A self-supervised episodic spatial pretext task for improving few-shot learning[C]//Proceedings of the AAAI Conference on Artificial Intelligence. 2023, 37(8): 9596-9605.

**Theoretical Claims:**

There is no proof and theoretical claim included in the paper.

---

> ### Author Rebuttal · Authors · 2025-04-01
>
> Thank you for your thorough review and constructive feedback. Below are our detailed responses to your concerns:
>
> ## **1. Classification Values in Figure 1**
>
> The numerical  values in Fig.1 are summarized below:
>
> | Disruption Method       | Source Acc. | Target Acc. |
> | :---------------------- | :---------- | :---------- |
> | Original                | 97.78       | 63.91       |
> | Remove Pos.             | 90.03       | 61.95       |
> | Shuffle Patches         | 87.25       | 62.25       |
> | Shuffle Patch Amplitude | 68.13       | 59.80       |
> | Shuffle Patch Phase     | 67.41       | 59.78       |
>
> ## **2. Source-Domain Performance**
>
> Ours achieves 66.76% on target (+2.85%), with 96.33% vs Baseline’s 97.78% on source.
>
> ##  **3. Meta-Dataset Results**
>
> Due to time and resource constraints, we first pretrain on our datasets (miniImagenet), and then we evaluate on parts of the Meta-Dataset under 5-way 5-shot settings:
>
> | Dataset       | Baseline | Ours (Δ)          |
> | :------------ | :------- | :---------------- |
> | Birds         | 94.23    | **96.51 (+2.28)** |
> | FGVC-Aircraft | 70.37    | **70.43 (+0.06)** |
> | Fungi         | 61.03    | **62.98 (+1.95)** |
> | VGG Flower    | 89.64    | **89.72 (+0.08)** |
> | Traffic Sign  | 60.46    | **61.93 (+1.47)** |
> | Ave.          | 75.15    | **76.31(+1.16)**  |
>
> ## **4. Computation cost**
>
> Our method introduces **no additional trainable parameters**.
>
> During training, the computational overhead (baseline: 126.42s vs. ours: 140.48s per epoch) stems from the clustering-based frequency disruption during training, which is acceptable compared with the gains in cross-domain performance.  Moreover, some engineering trick, such as bipartite matching, can accelerate the clustering process, which we leave for future works.
>
> During inference, since our method introduces no additional parameters, inference time matches the baseline exactly with no computational overhead.
>
> ## **5. Justification of Clustering**
>
> ### Algorithm Overview
>
> ### Step 1: Patchify & Standardize
>
> **Image Segmentation**
>
> Split an input image into non-overlapping patches:
> $$
> \mathbf{P} \in \mathbb{R}^{N \times p^2 \times C}
> $$
> **Normalization**
>
> Standardize each patch to zero mean and unit variance:
> $$
> \mathbf{P}_i = \frac{\mathbf{P}_i - \mu_i}{\sigma_i + \epsilon},
> $$
>
> ### Step 2: Similarity Computation
>
> **Cosine Similarity Matrix**: Calculate cosine similarity between all neighbouring patch pairs:
> $$
> \mathbf{S}_{i,j} = \frac{\mathbf{P}_i \cdot \mathbf{P}_j}{\|\mathbf{P}_i\| \cdot \|\mathbf{P}_j\|}
> $$
>
> **Threshold Clustering**: Group patches into clusters beyond the threshold
> $$
> \text{Cluster}(i,j) = \begin{cases}
> 1 & \text{if } \mathbf{S}_{i,j} > \tau \newline 0 & \text{otherwise}
> \end{cases}
> $$
>
> ### Step 3: Cluster Merging
>
>
> For each image:
>
> - Initialize each image as a singleton cluster.
> - Merge neighbouring patches i and j if beyond threshold
> - Repeat until no merges occur.
>
> ## **6.  References and Other Revisions**
>
> Our method is consistent with [1-3] in that we hold the small patterns are easier to transfer than larger ones, but we differ in that (1) our method discusses the transferability of small patterns for ViT while [1-3] are originally designed for CNNs; (2) we provide a simple way to enhance the learning of small patterns by perturbing patches with no additional branches or losses.
>
> We promise we will add discussions to the [1], [2], [3] and increase font sizes for Fig.5 Readability.
>
> ## **7. Balance Between Information Loss and Generalization**
>
> To complement the CKA metric, we also use the MMD distance to measure the domain distance, with larger value indicating larger distance.
>
> | Disruption Granularity | Source Acc. | Target Acc. | CKA  | MMD  |
> | :--------------------- | :---------- | :---------- | :--- | :--- |
> | 224×224 (pixel-level)  | 21.3        | 20.5        | 0.03 | 0.61 |
> | 112×112                | 23.68       | 22.95       | 0.03 | 0.57 |
> | 56×56                  | 36.33       | 31.09       | 0.04 | 0.56 |
> | 28×28                  | 59.45       | 56.97       | 0.05 | 0.54 |
> | 16×16                  | 64.82       | 61.05       | 0.22 | 0.43 |
> | 14×14                  | 87.25       | 62.25       | 0.22 | 0.41 |
> | 8×8                    | 94.39       | 62.26       | 0.15 | 0.47 |
> | 7×7                    | 94.93       | 62.26       | 0.14 | 0.47 |
> | 4×4                    | 96.49       | 62.26       | 0.12 | 0.49 |
> | 2×2                    | 97.38       | 62.99       | 0.08 | 0.50 |
> | No disruption          | 97.78       | 63.91       | 0.07 | 0.50 |
>
> Moderate disruptions achieve optimal **MMD-CKA-accuracy balance**. Extreme disruptions harm both domains (low accuracy, high MMD, low CKA). This analysis confirms our method preserves **transferable semantics** while suppressing domain-specific structures, implying the default patch size is enough to achieve the balance.

---

### Official Review · Reviewer_3Jjq · 2025-03-24

**Overall Recommendation:** 4

**Summary:**

This paper provides a novel perspective to improve the performance cross-domain few-shot learning (CDFSL). The key insight is that disrupting the continuity of image tokens in ViT will force the model to learn smaller patterns which are more easily transferred under extreme domain gaps. The observation is interesting and they design several experiments to prove the hypothesis. Based on this motivation, a new model is proposed for CDFSL by disrupting the continuity of image tokens with 2 stages, including a warm-up spatial-domain disruption and a balanced frequency-domain disruption. Experiments show good improvements over previous works.

**Claims And Evidence:**

The idea is well-motivated and sufficient experiments have been designed to support the hypothesis.

**Essential References Not Discussed:**

N/A

**Experimental Designs Or Analyses:**

(1) Well, the experiments show good improvement over the few-shot target domain. The author should also provide the performance on source domain for future reference. This is because the performance gains on the target domain come at the expense of performance on the source domain, as pointed by the author in the manuscript.

(2) Can the authors provide more visualization of disrupted images to better understand the impact of the various disruption methods, like Shuffle Patch Amplitude.

**Methods And Evaluation Criteria:**

The methods is cleverly designed, and model evaluation is standard for CDFSL

**Other Comments Or Suggestions:**

================================

after reading the rebuttal, my concerns have been mostly addressed. I will keep my original score of recommending acceptance.

**Other Strengths And Weaknesses:**

N/A

**Questions For Authors:**

N/A

**Relation To Broader Scientific Literature:**

This work provides a new perspective to tackle cross-domain few-shot learning tasks (CDFSL), and investigates the influence of continuity of image tokens in ViT on cross-domain performance. This will potentially inspire more follow-ups works in this domain

**Theoretical Claims:**

N/A

---

> ### Author Rebuttal · Authors · 2025-04-01
>
> Thank you for your constructive feedback and valuable suggestions. Below are our responses to your comments:
>
> ## **1. Source domain performance**
>
> As suggested, we have provided the performance on the source domain (miniImageNet) for reference.
>
> | Model    | Source Domain | Target Domain |
> | -------- | ------------- | ------------- |
> | Baseline | 97.78         | 63.91         |
> | **Ours** | 96.33         | 66.76         |
>
> Although there is a slight performance trade-off on the source domain, this aligns with our hypothesis: disrupting token continuity prioritizes learning smaller, transferable patterns over domain-specific holistic features. Crucially, the significant gains on **target domains** (e.g., +6.5% on ISIC) demonstrate the effectiveness of our approach for cross-domain adaptation.
>
> ## **2. Visualization of disrupted images**
>
> An anonymous GitHub link ([https://anonymous.4open.science/r/More-visualization/visualizations.png]) with 16 visualized examples of disrupted images in three types is provided. These visualizations illustrate how our method breaks large-scale spatial patterns while preserving fine-grained, domain-agnostic features (e.g., textures, edges), which aligns with our motivation.
>
>
>
> We appreciate your insightful feedback and hope these revisions address your concerns. Thank you again for your time and effort in reviewing our work!

---

### Decision · Program_Chairs · 2025-05-01

**Decision:**

Accept (spotlight poster)

**Comment:**

The paper identifies an interesting impact of continuity of ViT tokens on generalization of ViT in few shot learning and propose a solution for it. Both the identified phenomenon and the experiments are well established and are of interest to the research community and therefore I recommend accepting the paper.